# Seismic Performance of Story-Added Type Buildings Remodeled with Story Seismic Isolation Systems

**Moo-Won Hur**  **and Tae-Won Park \***

Department of Architectural Engineering, Dankook University, Yongin-si 16890, Korea; hmw01@nate.com
\* Correspondence: tw001@dankook.ac.kr; Tel.: +82-31-8005-3751

**Abstract:** Story-added type apartments have recently been introduced as an option to resolve the housing supply shortage in areas that are undergoing rapid industrialization and urbanization. However, the infrastructure of old apartment buildings (>20 years old) makes it difficult to introduce convenient facilities and recent technologies such as those involving the Internet of Things and augmented realities. Applying housing technologies to existing older apartments can increase housing supply and potentially address the aforementioned issues. However, story-added building remodeling increases the weight of existing buildings, necessitating seismic reinforcement, which is the major obstacle when performing vertical building extensions. This study presents methods for lowering seismic loads associated with vertical augmentation of buildings while improving the seismic performance. A model of a vertically extended building with three additional stories constructed on top of an existing 15-story apartment building was used. The applied seismic isolation system decreased the maximum response acceleration on top of the remodeled building by approximately 70% and 65% in the X-direction and Y-direction, respectively, while decreasing the base shear plane by approximately 30% in both the X- and Y-directions in comparison with forces on a non-seismically isolated building. These results demonstrate that the use of a seismic isolation system can significantly reduce seismic loads.

**Keywords:** remodeling; vertically story-added; isolation system; isolation period; maximum response acceleration; maximum response displacement



## 1. Background

Story-added type apartments have recently been introduced as an option to resolve housing supply shortage in areas that are undergoing rapid industrialization and urbanization. However, the infrastructure of old apartment buildings (>20 years old) makes it difficult to introduce convenient facilities and recent technologies such as those involving the Internet of Things and augmented realities. Several factors can inhibit the initiation of apartment reconstruction projects, including the following: (i) social issues associated with the rapid decline in residential quality; (ii) environmental disruption and waste of resources associated with apartment reconstruction; (iii) economic factors regarding a decrease in real estate value when buildings become taller.

Despite the demand for reconstructing residential apartments increasing, there is a panoply of adverse effects that can affect such reconstruction projects, including social issues arising from the overheated housing market, in which most buyers prefer new buildings, the lack of rental houses in the market, along with technical issues such as environmental destruction associated with the mass production of waste materials from demolished buildings, excessive carbon dioxide emissions owing to the increasing demand for large amounts of concrete, and the lack of natural aggregates for the reconstruction of apartments.

Furthermore, due to the continuing urbanization and industrialization of our society, there is a need to supply houses to meet the demand for housing. Thus, the government

has begun to recommend the remodeling of existing apartment buildings in order to embrace the economic and environmental benefits of remodeling and increase the supply of houses by approximately 15%. The government recommends the vertical extension of older buildings in order to simultaneously solve issues associated with the lack of houses and the deterioration of old apartments.

However, recommendations for vertical extensions are accompanied by concerns about increased lateral loads caused by building vertical extensions, as well as increases in the number of stories and total floor area. That is, an increase in the total weight of a building can result in increases of wind load and seismic load. This eventually necessitates reinforcement of such buildings to account for the increased load on the foundation and the augmentation of members against the lateral resistance caused by increases in wind load and seismic load (ASCE/SEI41-17 [1]). An increase in the vertical load can be compensated for by a design capable of bearing approximately 15% of the axial load applied to the cross-section of a building, whereas the lateral load may exceed the level of the designed load with building extensions due to the lower safety factor of a horizontal load. This necessitates a separate lateral reinforcement of the building. The cost of developing a separate lateral reinforcement option may be equal to or greater than the cost of a new construction, resulting in economic losses connected with a remodeling project (refer to Figure 1 and Table 1).

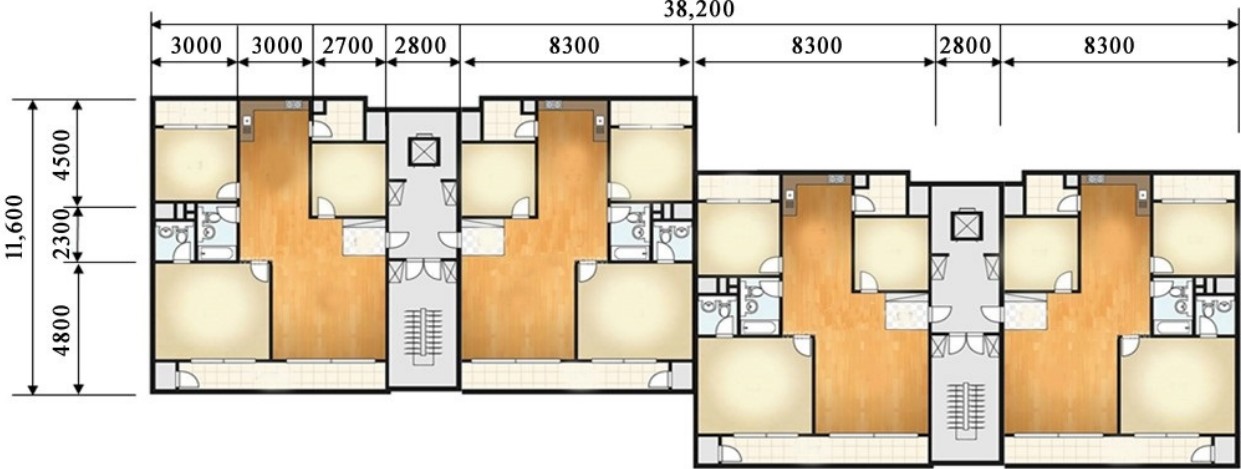

**Figure 1.** Floor plan of an apartment with an exclusive area of 85 m$^2$ (real apartment).

**Table 1.** Vertical stress before and after remodeling (apartment with an exclusive area of 85 m$^3$).

| Weight of Building | | Wall | Slab | Live Load (Including Floor Finish) | Total Load |
|---|---|---|---|---|---|
| Weight of the first floor (kN) | | 2700 | 1780 | 1980 | 6460 |
| Cross-sectional area of the wall (cm$^2$) | | | | 433,200 | |
| Before remodeling of the building (15 Stories) | Total weight of building (kN) | 40,500 | 26,700 | 29,700 | 96,900 |
| | Average vertical stress of the first floor (kN/cm$^2$) | | | 96,900/433,200 = 0.2237 | |
| After remodeling of the building (18 Stories) | Total weight of building (ton) | 46,080 | 32,760 | 37,440 | 115,380 |
| | Average vertical stress of the first floor (kN/cm$^2$) | | | 115,380/433,200 = 0.2663 | |
| Increment in vertical load | | | | 115,380 − 96,900 = 18,480 kN | |
| Increment in vertical stress | | | | 0.2237 → 0.2663 (Approximately 12.0% of Allowable Stress of the Material) | |

A seismic isolation system can be applied to cope with such problems. The application of such a system is advantageous as it can increase the safety of the existing building against earthquakes without the need to add any separate structural members, which may damage existing buildings more than conventional approaches. Xiaoyu Gu (2020) has conducted a study on MR elastomer (MRE)-based isolators and demonstrated their performances through a shake table experiment to assess a three-story shear building [2]. Antonello De Luca (2020) has examined issues associated with the evolution of an Italian style of seismic isolation [3]. He highlighted the advancements made during the last 30 years, which involved the transformation of early pioneering concepts into cutting-edge solutions. Furthermore, Mohammad Masoud Pourmasoud (2020) has developed a "Multi-Directional Seismic Isolation (MDSI)" system and demonstrated its performance through experiments [4].

Recent papers on seismic isolation systems have mostly reported the performance of seismic isolation by conducting pertinent experiments. Few papers have addressed the applicability of these systems.

In the present study, the effect of the application of a seismic isolation system was analyzed to provide basic data required to examine the applicability and development of the system for vertical extension remodeling of an old apartment building. The seismic isolation system can vary according to the number of added stories in the vertical extension and the seismic isolation period of the building. A seismic isolation system was then applied to the actual building. The resulting effects were examined based on the computed optimal seismic isolation period. In addition, the applicability of the seismic isolation system to few-story buildings used as schools, where the application of aseismic reinforcement has recently been increased, was assessed. In this manner, basic data required for the development of plans for the vertical expansion of old buildings through the application of seismic isolation systems are presented.

## 2. Seismic Isolation Systems for Old Building Remodeling by Adding Stories

A seismic isolation system is a practical option for protecting important buildings (data centers, hospitals, etc.) from powerful earthquakes and improving the aseismic performance of existing architecture. Recently, the number of designs that aim to exploit seismic isolation systems, especially designs for data centers according to client requirements, has been increasing. To reflect the reality of the situation, seismic isolation system requirements were added to ACI 318-19 [5].

Seismic isolation systems have been designed to be used for buildings such as residential welfare centers, apartments, composite buildings with both apartments and commercial facilities, data centers, computer centers, and so on. However, the number of newly installed seismic isolation systems is continuously declining due to rising frame structure building costs and extended construction duration [6–17].

Nonetheless, the application of seismic isolation systems to stories added to old apartment buildings as suggested in the present study is estimated to be feasible as it entails a shortened construction period and minimizes the amount of reinforcement needed to ensure the aseismic performance of existing buildings. The advantage of using a seismic isolation system in this manner is that it frees up space for floor plans or improves the design of floors of buildings that are above the seismic isolation system. In addition, aseismic reinforcement of an existing building can be minimized by reducing the load delivered to the existing lower structure of a building (refer to Figure 2).

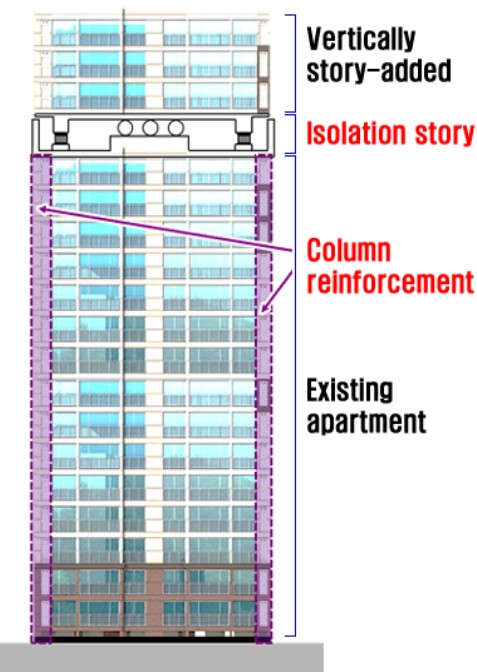

**Figure 2.** Concept of vertical story-added isolation.

## 3. Preliminary Analysis

### 3.1. Model of Preliminary Analysis

According to Roehl (1972), a single-span frame with a story height and span width 2 h was installed to an existing building to test effects of seismic isolation in the context of story addition [18]. Cross-sectional areas of all sections as well as the mass and stiffness distribution on each floor of the structure were considered to remain constant. Figure 3 shows a schematic illustration of the model. Based on the simplified formula generally used to express each floor of a building, the period of vibration of the building was set within the following range:

$$0.6T \leq T_1 \leq 1.4T \tag{1}$$

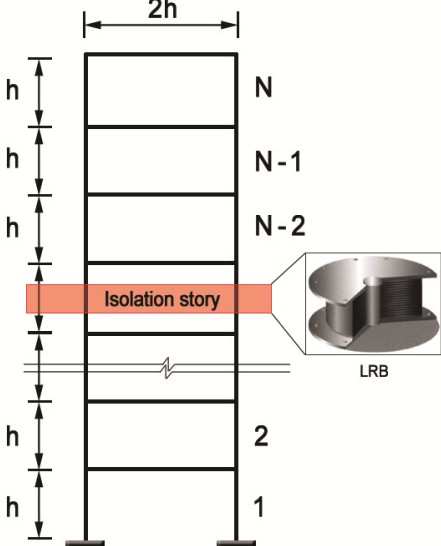

**Figure 3.** Analytical model and isolation device.

Here, T = 0.1 N, where N = the number of stories in the building (ASCE/SEI41-17).

The building used for the analysis has a total of 15 stories. The current regulatory standards allow adding up to three stories. Hysteretic characteristics used in the design of an actual seismic isolation device (bilinear model) were used for the present analysis. Taking into account the aforementioned characteristics and the scope of the research, the story was combined with three potential stories and three models of the seismic isolation period. The resultant nine models were analyzed. Regarding earthquake (EQ) vibrations, data from the El Centro EQ were used for the boundary nonlinear time history analysis. MIDAS Gen 2020 was used to model the frame elements of upper and lower structures, while the seismic isolation device was modelled with bilinear characteristics along the two in-plane directions. The two shear strain springs included in the lead rubber bearing isolator have characteristics of interrelated basal plasticity and independent linear elastic springs for variations of the remaining four degrees of freedom That is, a total of two seismic isolation devices were used. They were installed one by one on each column.

### 3.2. Results of Analysis

### 3.2.1. Effect of Seismic Isolation on One Story Added to a Building

As shown in Figure 4, the seismic isolation period of the addition of one story to an apartment building was compared with periods obtained from twice, three times, and four times the seismic isolation of a non-seismically isolated building.

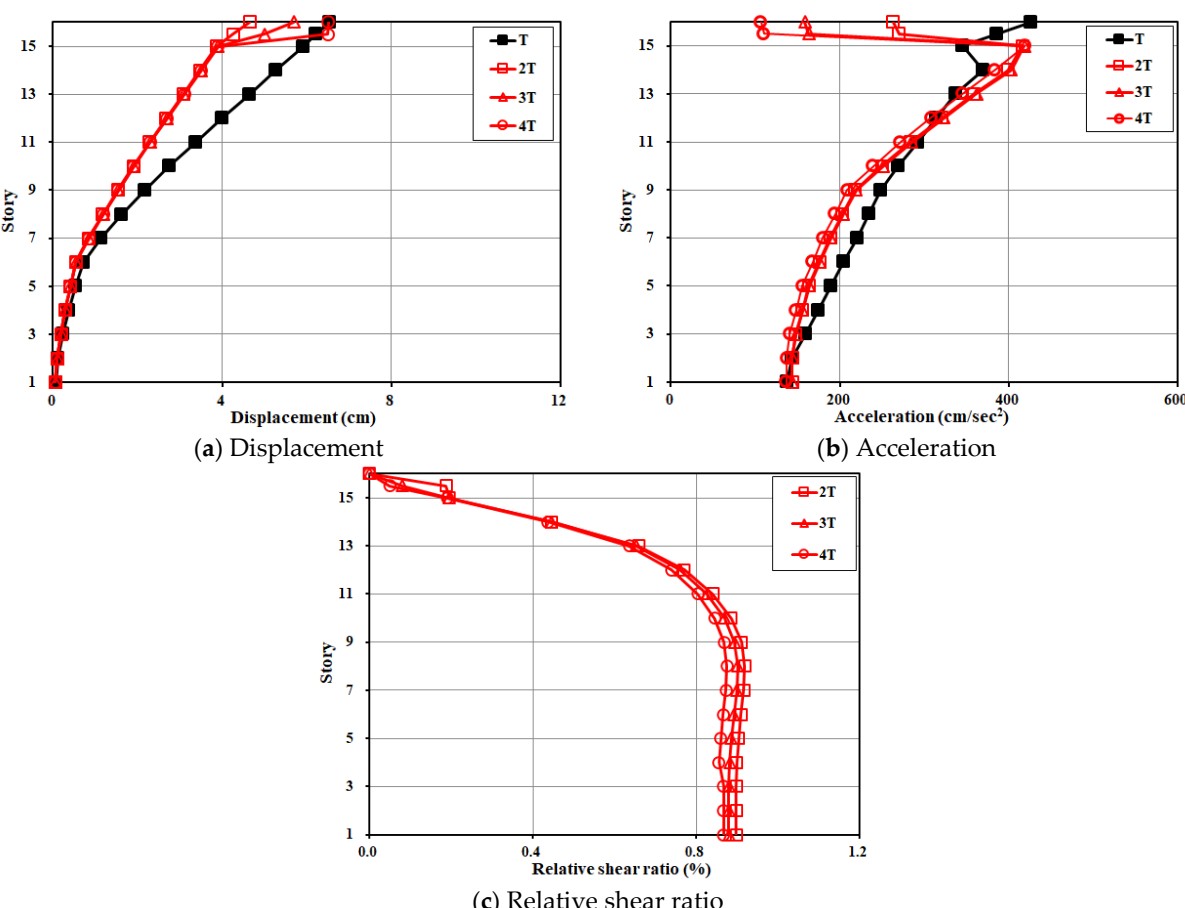

(**a**) Displacement                                                   (**b**) Acceleration

(**c**) Relative shear ratio

**Figure 4.** Results of vertical extension (one story-added).

In terms of the maximum response displacement and maximum response acceleration, the behavior of a rigid body did not appear in cases involving twice or three times the seismic isolation period, respectively. A behavior similar to that of a non-seismically isolated building appeared. However, displacement response and acceleration response appeared in a form similar to those of a rigid body at four times the seismic isolation period.

These results were attributed to the relatively lower weight of the story that was installed on the top of the building as the hysteretic characteristics of the seismic isolation device were similar to those of linear ones despite the identical period of seismic isolation resulting in a reduced seismic isolation effect.

Results of analysis of story-shearing force yielded a decrease of approximately 10% in all periods of seismic isolation compared to those of the non-seismically isolated building. In the event of a one-story addition, the above results indicate that more than four times the seismic isolation duration of a non-seismically isolated structure must be obtained for the upper part of the seismic isolation floor.

### 3.2.2. Effect of Seismic Isolation on Two Stories Added to a Building

As shown in Figure 5, the seismic isolation period of two stories added to an apartment building was compared with results obtained from twice, three times, and four times the periods of seismic isolation of the non-seismically isolated building.

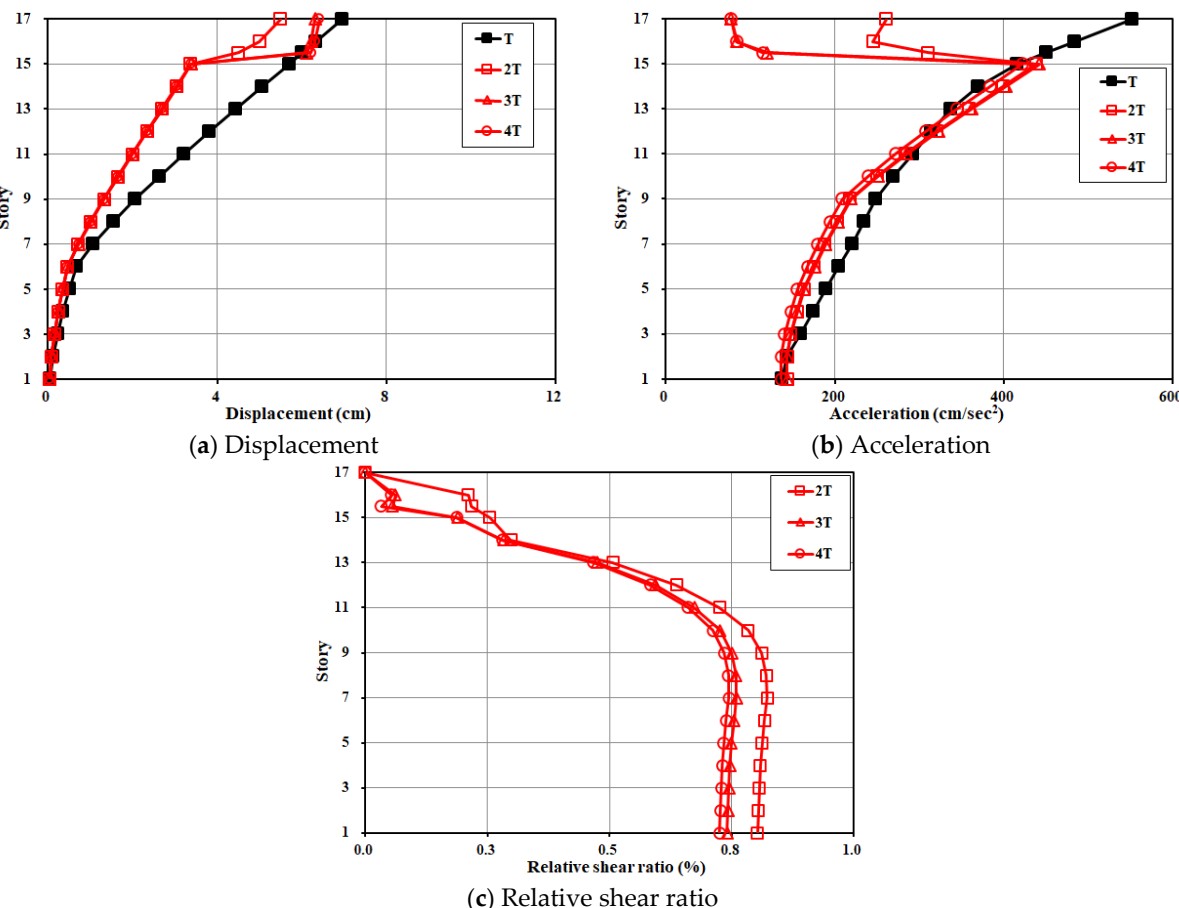

**Figure 5.** Results of vertical extension (two stories added).

The displacement response and acceleration response tended to increase with increasing height of the building in the absence of seismic isolation. However, those of seismically isolated buildings showed constant distributions of displacement response and acceleration response in the upper part of the building above the seismic isolation floor. In the case of the twice seismic isolation periods of the non-seismically isolated building, the displacement response did not exhibit rigid body behavior while the acceleration response increased. These results were attributed to the function of the installed seismic isolation device, which was insufficient. Results of analyses on story-shearing force showed values of approximately 20–25% at all periods of seismic isolation. The above results indicate that a seismic isolation period of at least three times greater than that of the non-seismically

isolated building needs to be secured for the upper part of the seismic isolation floor in the case of adding two stories to a building.

### 3.2.3. Effect of Seismic Isolation on Three Stories Added to a Building

As shown in Figure 6, the seismic isolation period of three stories added to an apartment building was compared with results obtained from twice, three times, and four times the period of seismic isolation of a non-seismically isolated building.

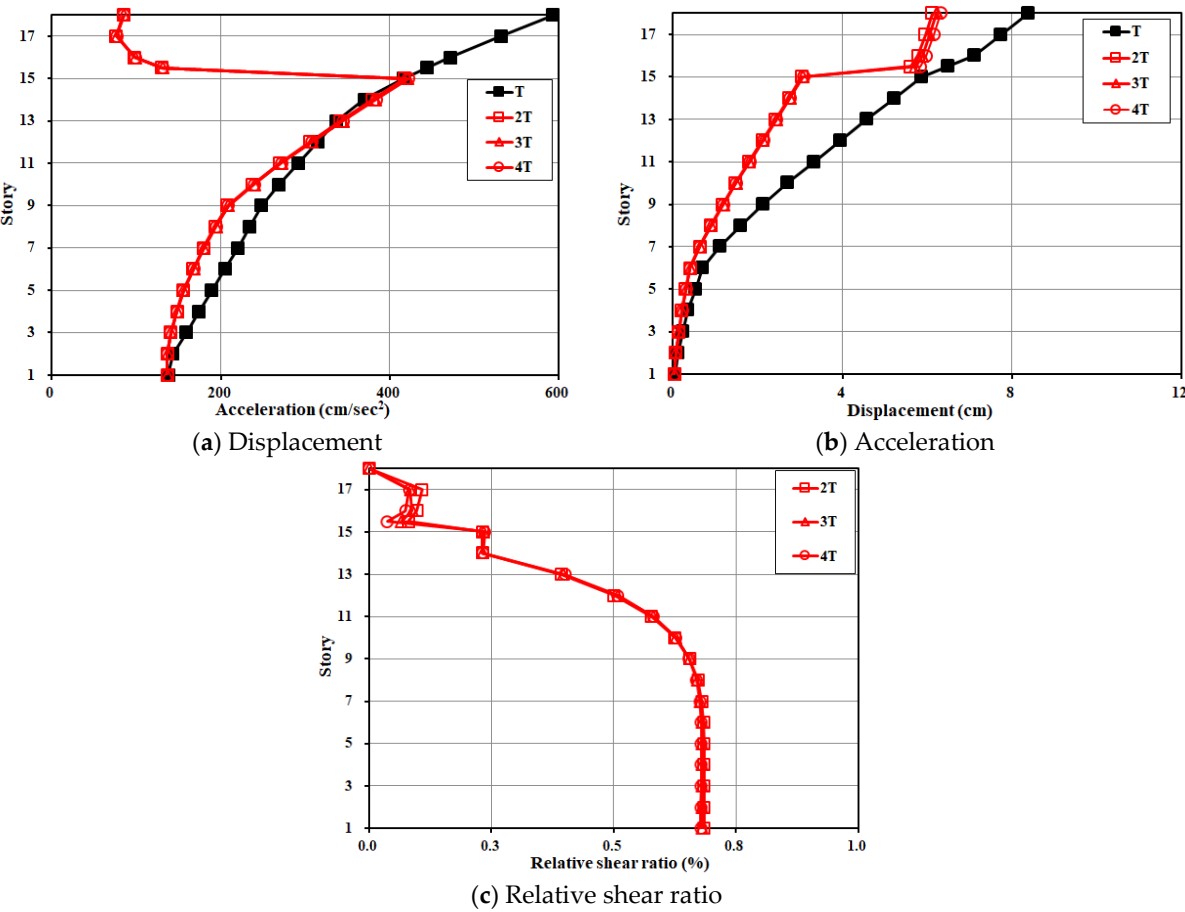

(**a**) Displacement          (**b**) Acceleration

(**c**) Relative shear ratio

**Figure 6.** Results of vertical extension (three stories added).

As shown in the figure, the non-seismically isolated building exhibited increasing values of maximum response displacement and maximum response acceleration in accordance with the increasing height of the building due to the addition of stories, whereas the seismically isolated building showed both the maximum response displacement and maximum response acceleration of the upper part above the seismic isolation floor of the building, which were similar to each other. In addition, the maximum response displacement, the maximum response acceleration, and effects of the reduction in story-shearing force appeared to be insignificant.

Regarding the effect of story-shearing force, a reduction in shearing force of approximately 30% at the base plane commonly appeared in cases of twice, three times, and four times the seismic isolation period of the seismically isolated building compared to that of the non-seismically isolated building. The decrease in shearing force implied a decrease in weight on the lower structure of the building, which indicated that materials needed for aseismic reinforcement of the lower parts of the structure could be saved, thus reducing economic concerns. In the case of a three-story addition, the above results revealed that the intended seismic isolation effect would be possible with the seismic isolation period of the

upper part of the seismic isolation floor being more than twice that of the non-seismically isolated building.

## 4. Evaluation of the Applicability of a Seismic Isolation System for Remodeling an Apartment Building with Stories Added

### 4.1. Research Model

The apartment building selected for the research model in the present study was built in the early 1990s. It was located in Seongnam City, Korea. It was a 15-story apartment building made from an ordinary reinforced-concrete shear-wall structure. This building allowed for the addition of up to three stories. Regarding the strength of its members, the compressive strength of the reinforced concrete was 21 MPa, while the yield strength of the reinforcing bar was 400 MPa. The period of the apartment building before the application of a seismic isolation system was 1.10 sec. The fundamental design wind load ($V_O$) of the apartment building was 26.0 m/sec and the exposure to wind was exposure B (ASCE/SEI41-17). Regarding seismic loading, coefficients of locality (A), subgrade reaction, and importance ($I_W$) were 0.22 and 1.5, respectively. The site condition of the measured site was bedrock ($V_{s,30} > 760$ m/s). Figure 7 illustrates the plan and elevation of the apartment building. Figure 8 shows the results of the elastic analysis review. All members were found to have sufficient strength.

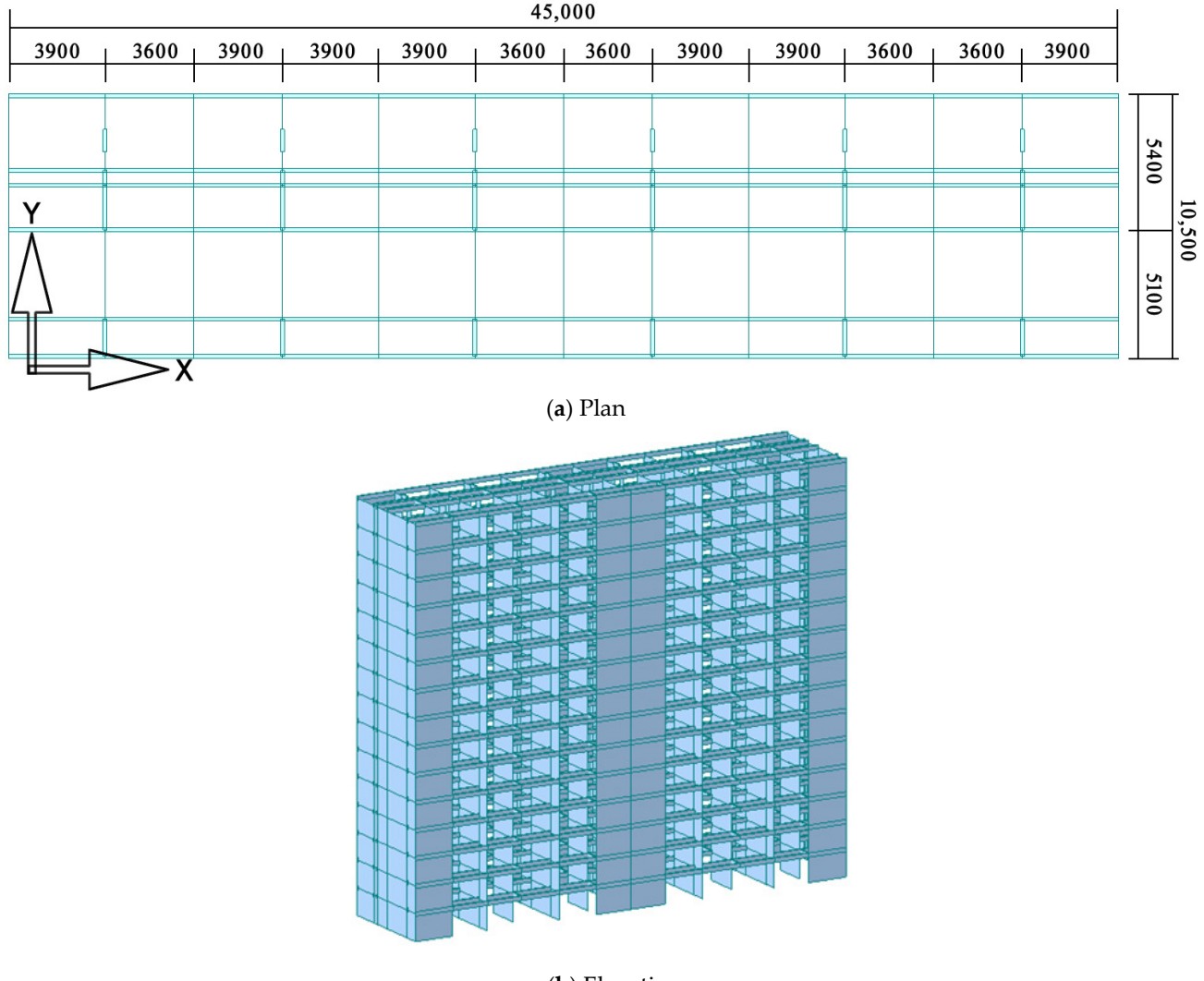

(**a**) Plan

(**b**) Elevation

**Figure 7.** Model of a 15-story apartment building.

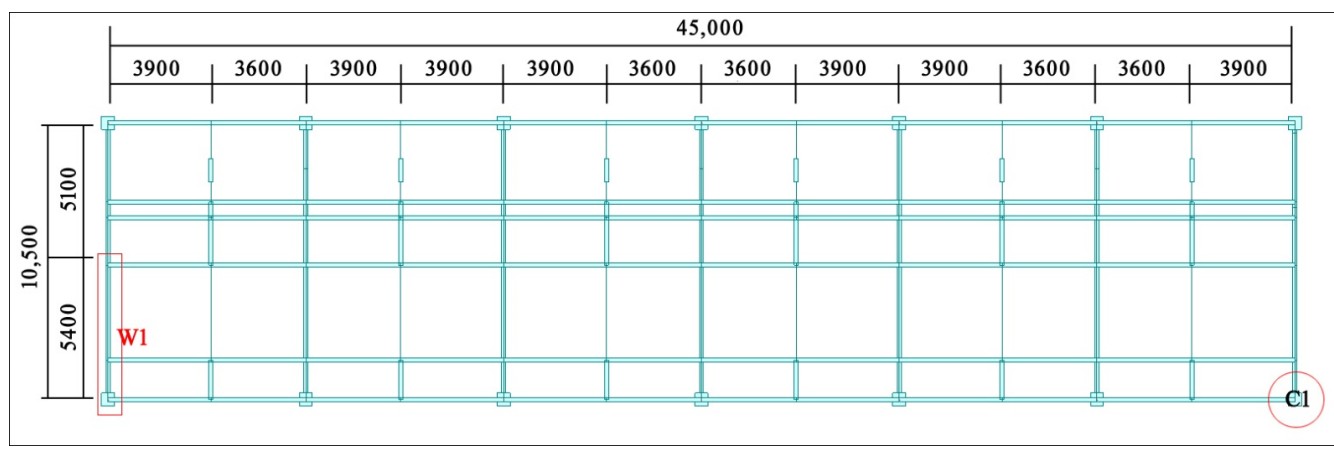

(**a**) Review member location

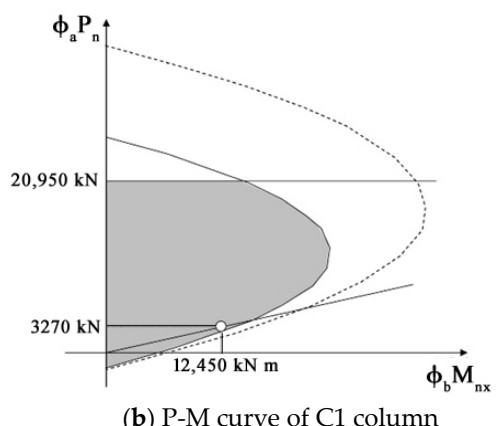

(**b**) P-M curve of C1 column

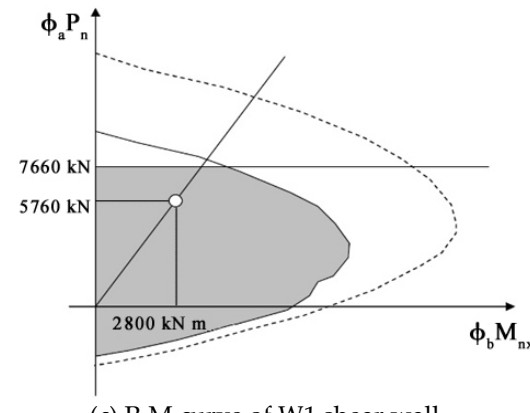

(**c**) P-M curve of W1 shear wall

**Figure 8.** Elasticity analysis results.

### 4.2. Wind and Seismic Loads

Lateral loads applied to a building can be broadly classified into wind loads and seismic loads. A seismic isolation device should always remain elastic when subjected to a wind load. A seismic isolation device that works with the application of a wind load may trigger issues related to serviceability. Thus, the wind load that may be felt by the apartment building needs to be measured and reflected in the design of the seismic isolation device. Table 2 lists the fundamental dynamic characteristics of the building before the addition of the story. Table 3 presents the wind load and seismic load applied to the building.

**Table 2.** Mode shapes of a 15-story apartment building.

| Mode No. | Period (s) | X-Direction Mass (%) | Y-Direction Mass (%) | Z-Direction Mass (%) |
|---|---|---|---|---|
| 1 | 1.10 | 66.0 | 0.0 | 0.0 |
| 2 | 0.50 | 0.0 | 65.1 | 0.1 |
| 3 | 0.12 | 0.1 | 0.1 | 65.3 |
| | Mode 1 | | Mode 2 | Mode 3 |

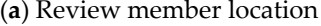



**Table 3.** Wind load and earthquake load of a 15-story apartment building.

| Story | Story Level (m) | Weight (kN) | Wind (kN) | | Earthquake (kN) | |
|---|---|---|---|---|---|---|
| | | | Force | Story Shear | Force | Story Shear |
| Roof | 39.0 | 3702.3 | 12.2 | 0.0 | 839.0 | 0.0 |
| 15F | 36.4 | 4334.3 | 24.3 | 12.2 | 908.4 | 839.0 |
| 14F | 33.8 | 4334.3 | 23.9 | 36.4 | 835.3 | 1747.5 |
| 13F | 31.2 | 4334.3 | 23.3 | 60.3 | 762.9 | 2582.8 |
| 12F | 28.6 | 4334.3 | 22.8 | 83.7 | 691.4 | 3345.8 |
| 11F | 26.0 | 4334.3 | 22.2 | 106.4 | 620.6 | 4037.1 |
| 10F | 23.4 | 4334.3 | 21.6 | 128.6 | 550.8 | 4657.7 |
| 9F | 20.8 | 4334.3 | 20.9 | 150.2 | 482.1 | 5208.6 |
| 8F | 18.2 | 4334.3 | 20.2 | 171.1 | 414.4 | 5690.6 |
| 7F | 15.6 | 4334.3 | 19.4 | 191.3 | 348.0 | 6105.0 |
| 6F | 13.0 | 4334.3 | 18.9 | 210.7 | 283.1 | 6453.1 |
| 5F | 10.4 | 4334.3 | 18.8 | 229.7 | 219.9 | 6736.2 |
| 4F | 7.8 | 4334.3 | 18.8 | 248.5 | 158.8 | 6956.1 |
| 3F | 5.2 | 4334.3 | 18.8 | 267.4 | 100.3 | 7114.8 |
| 2F | 2.6 | 4334.3 | 18.8 | 286.2 | 45.8 | 7215.2 |
| 1F | 0.0 | - | - | 305.1 | - | 7260.9 |

*4.3. Design and Arrangement of the Seismic Isolation Device*

4.3.1. Target Period of Seismic Isolation

The effective period of a seismically isolated building can be determined using the strain characteristics of the seismic isolation system. As mentioned previously, the effective period of seismic isolation at the design displacement of the three-story addition was set at 2.0 times ($\approx$2.0 s).

4.3.2. Effective Rigidity of the Seismic Isolation System

The effective period of a seismic isolation structure at the design displacement ($D_D$) can be determined using the strain characteristics of the seismic isolation system as expressed in the following Equation (2). By exploiting the effective period, the strain characteristics of the seismic isolation system that are needed to satisfy the targeted period of seismic isolation can be determined.

$$T_D = 2\pi\sqrt{\frac{W}{K_D \times g}} \tag{2}$$

Here, W denotes the effective weight of the upper part of the building (=13,002 kN) and $K_D$ represents the effective stiffness of the seismic isolation system at the design displacement.

$$K_D = \frac{4\pi^2 \times 13,002,000}{4 \times 9800} = 13.08 \text{ kN/mm}$$

4.3.3. Design of Seismic Isolation Device and Arrangement of Seismic Isolation System

According to the above conditions, lead rubber bearings (LRB) and natural laminated rubber supports were used as components of the seismic isolation system. The system consisted of a total of 15 seismic isolation devices. Specifications of the seismic isolation device and its hysteretic characteristics are presented in Table 4 and Figure 9, respectively. Figure 10 illustrates the position of the column reinforcement and the installation positions of seismic isolation devices. A total of 14 columns were added to the exterior of the building. The column size was 500 mm × 500 mm. The dimensions of the main bars and the hoop bars were 10-D22 and D10@200, respectively. The compressive strength of the concrete was 24 MPa and the yield strength of steel bars was 400 MPa.

**Table 4.** Details of isolation bearing.

|  | LRB (Lead Rubber Bearing) | RB (Rubber Bearing) |
|---|---|---|
| Outer diameter (D, mm) | 500 | 400 |
| Lead bar diameter ($D_i$, mm) | 90 | 15 |
| Rubber thickness (mm) | 4.0 | 3.2 |
| No. of rubber layer | 25 | 25 |
| Total rubber thickness (mm) | 100 | 80 |
| Steel plate thickness (mm) | 3.2 | 3.2 |
| First Shape factor S1 | 31.3 | 30.1 |
| Second Shape factor S2 | 5.0 | 5.0 |
| Lateral stiffness (kN/mm) | 0.9 | - |

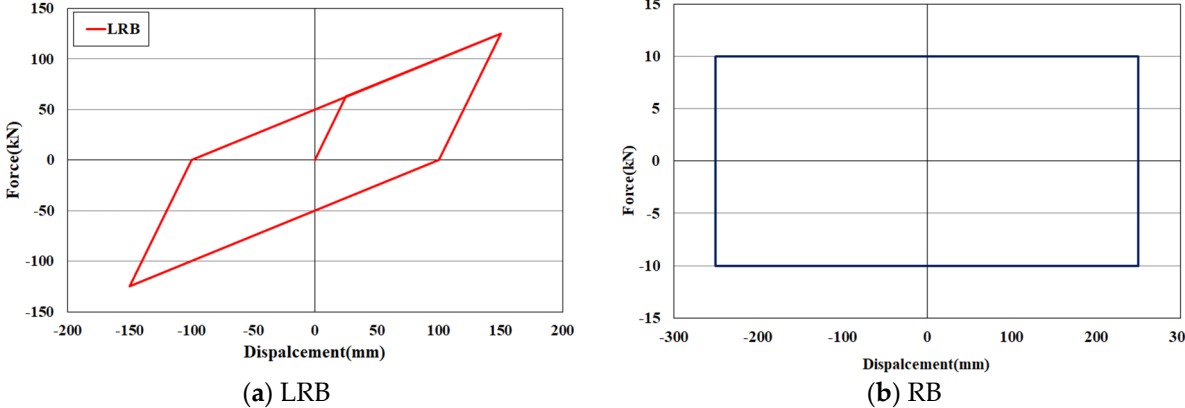

(**a**) LRB                                                                (**b**) RB

**Figure 9.** Hysteretic characteristics of isolation bearing.

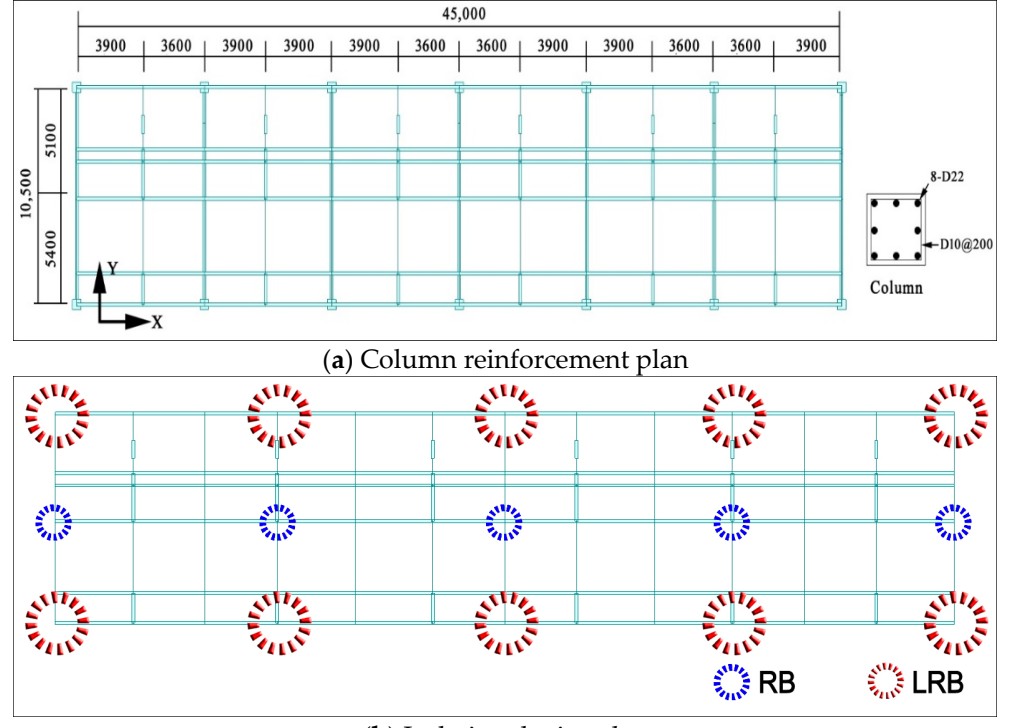

(**a**) Column reinforcement plan

(**b**) Isolation device plan

**Figure 10.** Reinforcement plan.

4.3.4. Examination of Wind Load

The seismic isolation floor of a building with a three-story addition stood against the wind load without yielding, resulting in a displacement of 3.5 mm in both the X- and Y-directions.

$$\text{Displacement of Seismic Isolation Floor} = 3.5 \text{ mm} \ < \ WD_y = 25 \text{ mm}$$

(Yield Displacement of Seismic Isolation Floor).

4.3.5. Calculation of Minimum Horizontal Displacement

The seismic isolation system should be designed according to Equation (3) to resist the minimum horizontal displacement ($D_D$) generated along the horizontal main axial direction of the structure. The seismic isolation device designed in the present study satisfied the $D_D$.

$$D_D \ = \ \frac{g \ \times \ S_{D1} \ \times \ T_D}{4\pi^2 \ B_D} \tag{3}$$

$$D_D = 164.6 \text{ mm} \ < \ \frac{D}{2} \ = \ \frac{500}{2} = 250 \text{ mm}$$

Here, g: gravitational acceleration, design displacement.

$S_{D1}$: Attenuation of design 5% with 1-s period, a variable regarding the acceleration of the spectrum.

$T_D$: Effective period of a seismically isolated structure at the design displacement of the direction considered (=2.0 s).

$B_D$: Numerical coefficient associated with the effective attenuation ($\beta_D$) of the seismic isolation system. In the present case, it was set at 1.5 by assuming 20% of effective attenuation (Table 5).

**Table 5.** Damping coefficient, $\beta_D$ or $\beta_M$ (data from ASCE/SEI41-17).

| Effective Damping, $\beta_D$ or $\beta_M$ (Percentage of Critical) | $B_D$ or $B_M$ Factor |
|---|---|
| $\leq 2$ | 0.8 |
| 5 | 1.0 |
| 10 | 1.2 |
| 20 | 1.5 |
| 30 | 1.7 |

*4.4. Incident Earthquake Vibration*

The incident earthquake vibration used in the present study was selected through a site response analysis. Earthquake measurements of bedrock collected in the "PEER Ground Motion Database" (USA) were screened and used for site response analyses. For screening purposes, ground conditions comprising bedrock (over $V_{s,30} > 760$ m/s) and the scale factor of the magnitude of the earthquake ranging from approximately 0.3 to 3.0 were exploited to select 40 records of earthquakes that had occurred in the bedrock. Site response analyses were conducted for these 40 records, from which a set of seven earthquake waves satisfying the limitations of the standards (90% of the value of 1.3 times of the design response spectrum) and exhibiting the design response spectrum with less deviation was constructed via an optimization algorithm [19].

Table 6 presents the seven incident earthquake vibrations selected through the above procedure. Figures 11 and 12 show the response spectrum of the "design-based earthquake (DBE)" of earthquake waves used for analysis and the acceleration time history of each earthquake wave, respectively. "MIDAS GEN 2021" was used for these analyses.

**Table 6.** Ground motions.

| No. | Event | Country | Year | Station | M | Vs, 30 |
|---|---|---|---|---|---|---|
| 1 | Chi-Chi | Taiwan | 1999 | HWA003 | 7.6 | 789 |
| 2 | Chi-Chi | Taiwan | 1999 | TAP067 | 7.6 | 808 |
| 3 | Chi-Chi | Taiwan | 1999 | ILA015 | 7.6 | 783 |
| 4 | Chi-Chi-06 | Taiwan | 1999 | HWA003 | 6.3 | 1526 |
| 5 | Loma Prieta | USA | 1989 | Piedmont Jr High School Grounds | 6.9 | 895 |
| 6 | Campano-Lucano | Italy | 1980 | Bisaccia | 6.9 | 958 |
| 7 | Loma Prieta | USA | 1989 | SF-Pacific Heights | 6.9 | 1250 |

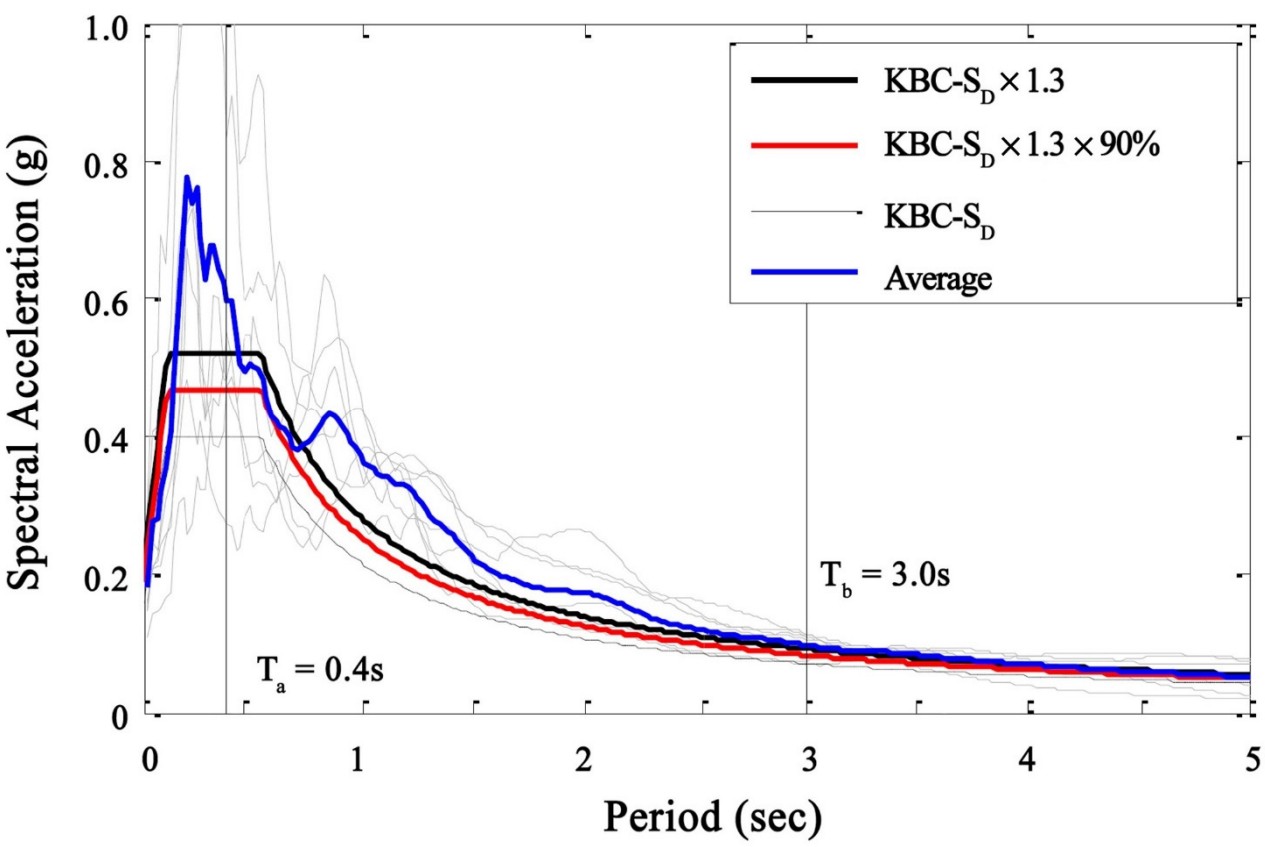

**Figure 11.** Frequency data of earthquakes used for the analysis (DBE).

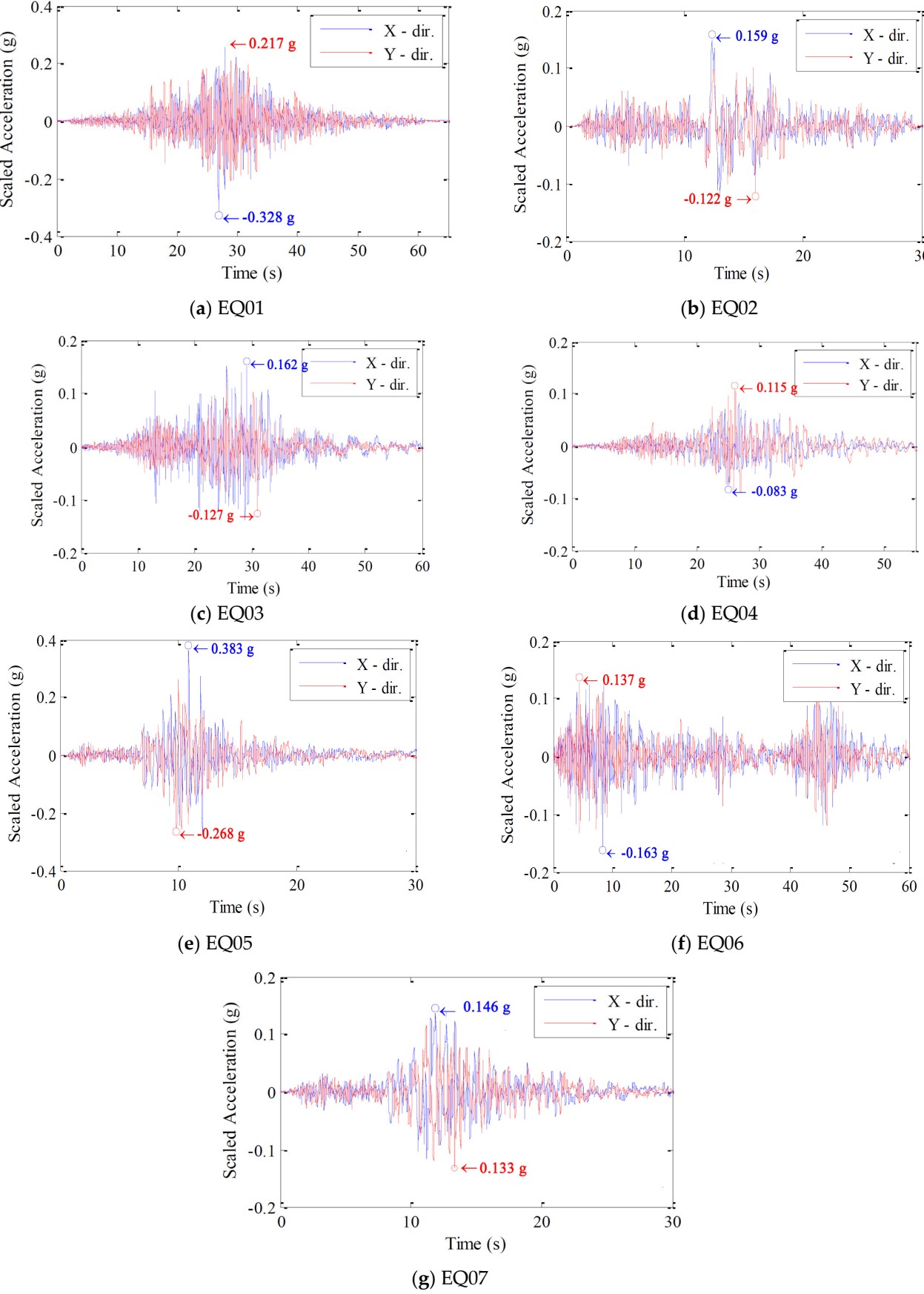

**Figure 12.** Earthquake waves.

### 4.5. Results

Table 7 lists the mean values obtained from the boundary nonlinear dynamic analyses of a building with a three-story addition with a seismic isolation system compared to those obtained from a building with a three-story addition without a seismic isolation system. These values are schematically illustrated in Figures 13–16. The maximum response displacement study indicated that the building with a seismic isolation system experienced significant displacement solely on the seismically isolated floor. In the absence of a seismic isolation system, the displacement response increased in proportion to the height of the building. In the presence of a seismic isolation system, the upper structure showed no difference in floor displacement, similar to the behavior of a rigid body. Here, the maximum displacement of the seismically isolated floor was found to remain within an allowable range of 205 mm.

**Table 7.** Non-linear dynamic analysis results.

| | Displacement (mm) | | | | Acceleration (mm/s$^2$) | | | | Shear (kN) | | | |
|---|---|---|---|---|---|---|---|---|---|---|---|---|
| **Story** | **Without Isolation** | | **With Isolation** | | **Without Isolation** | | **With Isolation** | | **Without Isolation** | | **With Isolation** | |
| | **X-Dir** | **Y-Dir** | **X-Dir** | **Y-Dir** | **X-Dir** | **Y-Dir** | **X-Dir** | **Y-Dir** | **X-Dir** | **Y-Dir** | **X-Dir** | **Y-Dir** |
| 19 | 99 | 45 | 99 | 63 | 3873 | 4893 | 1247 | 1722 | 0 | 0 | 0 | 0 |
| 18 | 92 | 42 | 98 | 63 | 2969 | 4337 | 1253 | 1718 | 1470 | 1885 | 557 | 664 |
| 17 | 85 | 39 | 98 | 62 | 2240 | 3810 | 1238 | 1727 | 2749 | 3825 | 1002 | 1086 |
| 16.5 | 81 | 37 | 97 | 62 | 2058 | 3565 | 1245 | 1710 | 3186 | 4662 | 1284 | 1387 |
| 16 | 78 | 36 | 55 | 22 | 1875 | 3320 | 2616 | 3844 | 3622 | 5499 | 2059 | 2031 |
| 15 | 71 | 32 | 50 | 20 | 1726 | 2998 | 2013 | 3315 | 4190 | 6904 | 2230 | 2454 |
| 14 | 64 | 29 | 46 | 18 | 1855 | 2832 | 1426 | 2949 | 4518 | 8061 | 2983 | 3586 |
| 13 | 57 | 26 | 41 | 16 | 2102 | 2702 | 1017 | 2702 | 4887 | 8991 | 3588 | 4539 |
| 12 | 50 | 23 | 36 | 14 | 2254 | 2648 | 1152 | 2522 | 5170 | 9756 | 3953 | 5324 |
| 11 | 44 | 20 | 31 | 13 | 2414 | 2674 | 1484 | 2395 | 5368 | 10,389 | 4122 | 6053 |
| 10 | 37 | 17 | 26 | 11 | 2550 | 2655 | 1769 | 2348 | 5439 | 10,966 | 4150 | 6747 |
| 9 | 31 | 14 | 22 | 9 | 2631 | 2598 | 1966 | 2292 | 5567 | 11,542 | 4044 | 7369 |
| 8 | 25 | 11 | 18 | 7 | 2635 | 2466 | 2058 | 2193 | 5964 | 12,302 | 3909 | 8034 |
| 7 | 20 | 9 | 14 | 6 | 2534 | 2287 | 2017 | 2062 | 6387 | 13,095 | 4144 | 8751 |
| 6 | 15 | 7 | 11 | 4 | 2357 | 2056 | 1860 | 1882 | 6826 | 13,791 | 4575 | 9469 |
| 5 | 10 | 5 | 7 | 3 | 2088 | 1815 | 1665 | 1723 | 7225 | 14,373 | 5009 | 10,098 |
| 4 | 6 | 3 | 5 | 2 | 1794 | 1599 | 1481 | 1579 | 7530 | 14,830 | 5413 | 10,602 |
| 3 | 3 | 1 | 2 | 1 | 1553 | 1460 | 1402 | 1474 | 7794 | 15,151 | 5743 | 10,955 |
| 2 | 1 | 1 | 1 | 0 | 1418 | 1404 | 1412 | 1437 | 8065 | 15,334 | 5933 | 11,160 |
| 1 | 0 | 0 | 0 | 0 | 1456 | 1456 | 1456 | 1456 | 8196 | 15,405 | 6005 | 11,240 |

Similar to the maximum displacement response data, data on maximum response acceleration revealed that a building without a seismic isolation system showed an increased response acceleration as one approached the upper floors. The reaction acceleration of stories above the seismic isolation floor of a building with a seismic isolation system appeared to be almost constantly distributed, leading to the expectation of enhanced structural system behavior. This could help prevent implosion in the event of an earthquake.

As shown in Table 6, the maximum response acceleration of a seismically isolated building was much lower than that of a non-seismically isolated building. Decreases of approximately 70% in the X-direction and 65% in the Y-direction were noted for floors affected by the simulated earthquake. Such a decline in acceleration signified a decrease in the shearing force of the upper structure. As a result, aseismic safety can be secured rather easily with the application of a seismic isolation system.

When subjected to a simulated earthquake, shearing forces in both the X- and Y-directions decreased by approximately 30% compared to their corresponding values in the non-seismically isolated building. The decrease in story-shearing force caused by the application of a seismic isolation system indicated a reduction in working load, which in

turn decreased the materials required for aseismic reinforcement. Altogether, a reduction in the size of members to be installed in stories to be added could ensure the economic viability of the story being added. Ultimately, reducing the size of members to be installed in stories might help ensure economic viability.

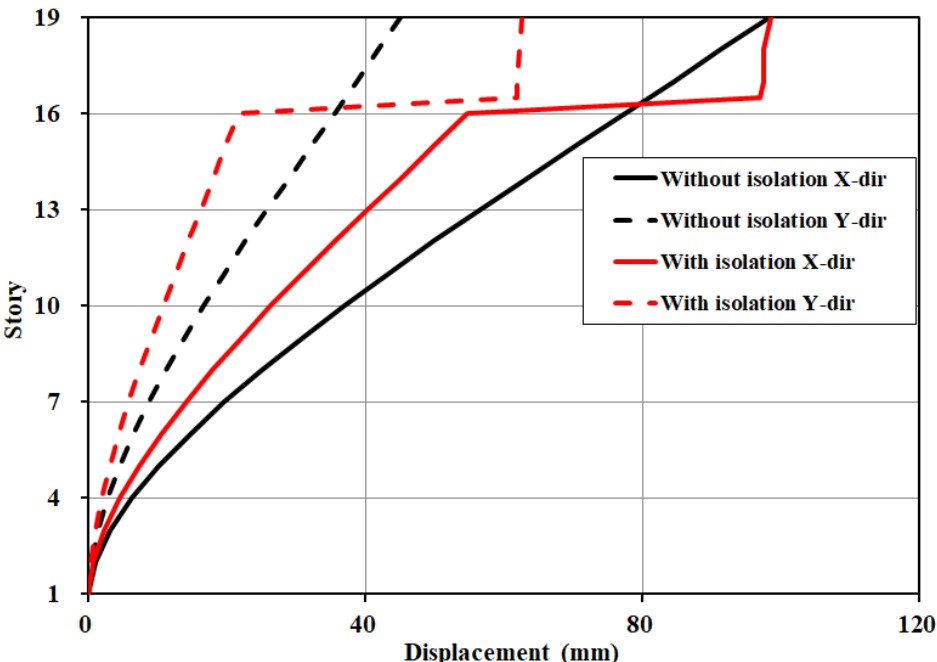

**Figure 13.** Maximum response displacement results.

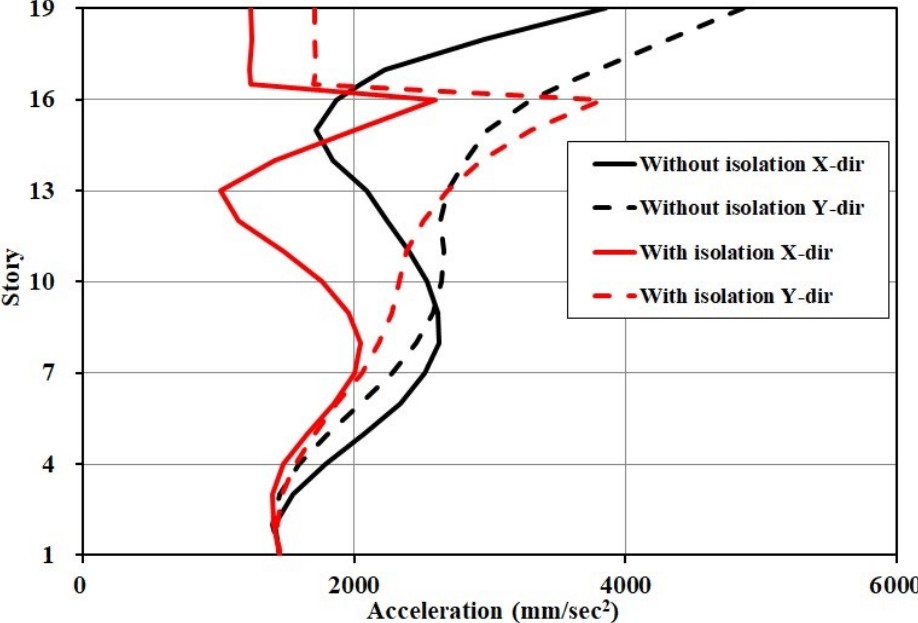

**Figure 14.** Maximum response acceleration results.

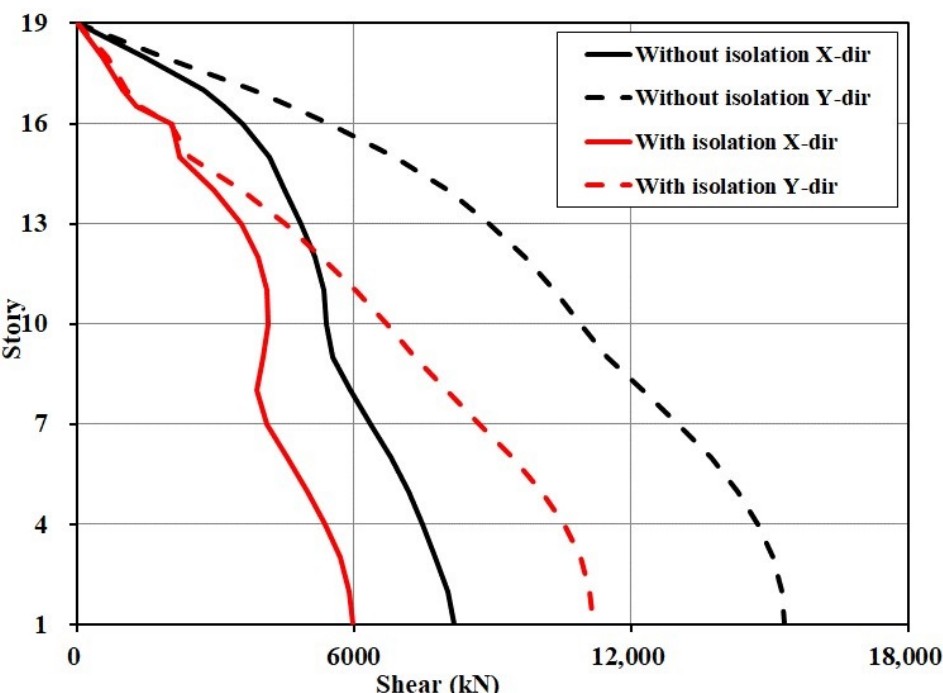

**Figure 15.** Story shear results.

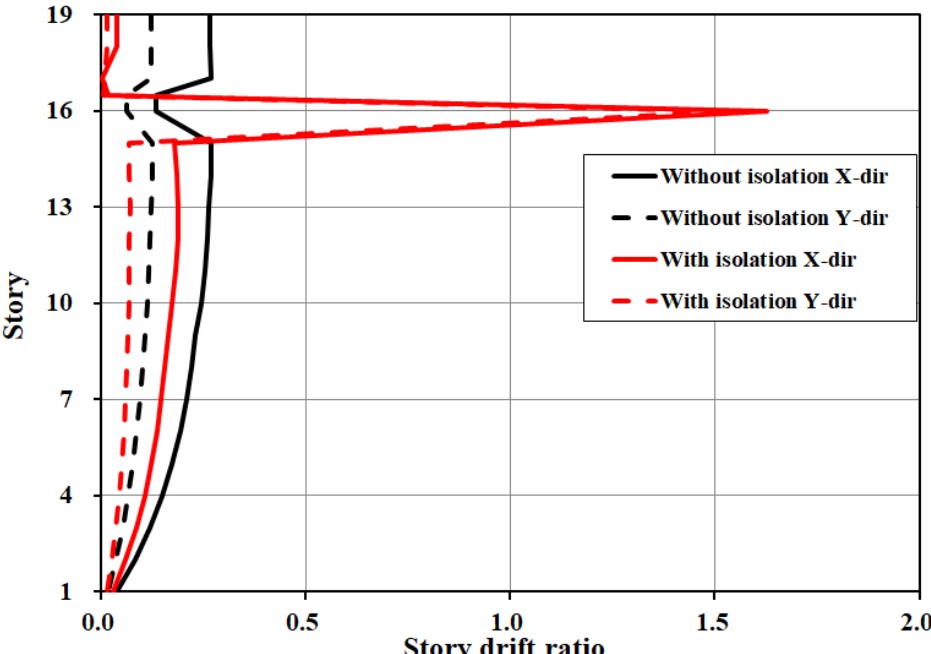

**Figure 16.** Inter-story drift.

## 5. Conclusions

The present study assessed how the number of stories affected a seismic isolation system. We measured the time of seismic isolation in order to provide basic data necessary for planning and examination processes related to applying seismic isolation systems to old apartment buildings with a story added. Based on the calculations of the optimal period of seismic isolation, the seismic isolation system was applied to an old apartment building that was remodeled by adding three stories. Based on the above results, the applicability of a seismic isolation system for remodeling involving the addition of a story to a few-story

Rahmen frame building was evaluated. The following conclusions could be drawn from the above analyses:

(1) The optimal period of seismic isolation according to the story added to each story of a building was examined by exploiting dynamic characteristics of the frame of the building. Findings revealed that adding a three-story (or a two-story) to a seismically isolated building with a period of seismic isolation that was more than twice (or thrice) as long as that of a non-seismically isolated counterpart might produce a suitable seismic isolation effect. In the case of the addition of one story, the sufficient seismic isolation effect was realized with a period of seismic isolation that was more than four times that of the non-seismically isolated building.

(2) The seismic isolation system was applied to a building with a three-story addition. The maximum response acceleration at the top of the seismically isolated building showed approximately 70% and 65% decreases in X- and Y-directions, respectively, compared to those of the non-seismically isolated building. Furthermore, regarding the shearing force at the base plane, the maximum response acceleration showed decreases of approximately 30% in both the X- and Y-directions. The decrease in story-shearing force implied a decrease in the working load on the building's foundation, which also suggested a reduction in materials needed for members involved in aseismic reinforcement.

(3) The effective period of seismic isolation for a three-story addition can be obtained by setting the target period of seismic isolation to be twice that of a non-seismically isolated building. Thus, for the case of the Rahmen frame building presented in a previous study (Hur, 2010), a period of seismic isolation that is more than 2.5 times greater than the innate vibration period of the upper structure, together with a target period of seismic isolation exceeding two seconds, is suggested for the design of stories added to attain the intended valid seismic isolation effect.

**Author Contributions:** Conceptualization, M.-W.H. and T.-W.P.; methodology, M.-W.H. and T.-W.P.; validation, M.-W.H. and T.-W.P.; formal analysis, M.-W.H. and T.-W.P.; investigation, M.-W.H.; analysis, M.-W.H.; resources, M.-W.H. and T.-W.P.; data curation, M.-W.H. and T.-W.P.; writing—original draft preparation, M.-W.H.; writing—review and editing, M.-W.H. and T.-W.P.; visualization, M.-W.H. and T.-W.P.; supervision, T.-W.P.; project administration, M.-W.H.; project administration, M.-W.H. and T.-W.P. All authors have read and agreed to the published version of the manuscript.

**Funding:** The present research was supported by a research fund of Dankook University in 2019.

**Conflicts of Interest:** The authors have no conflict of interest regarding the publication of this paper to disclose.

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
