# Peer review of "Seismic Performance of Story-Added Type Buildings Remodeled with Story Seismic Isolation Systems"

_buildings, doi:10.3390/buildings12030270_

Round 1

Reviewer 1 Report

The present manuscript studies the seismic behavior of existing buildings on which additional stories have been added and check the way for reducing the effects of the added stories. The authors propose the isolation of the added stories from the existing structure. They study and actual 15- story building and the resulting effects are presented. The authors have studied the addition of potential three additional stories, two stories or one story and three isolation periods. The building was subject to 7 earthquake ground motions and nonlinear analyses were performed.

In general, the paper must be clearly rewritten. The number of ground motions should clearly be presented. Also the authors should notice if the building is symmetric or not. The analyses were performed for uni or bidirectional ground motions? In general, the paper presents many drawbacks some of them are the following:

  • The abstract is not clear enough and is too long. It must be written again.
  • Page 3, line 117: width 2h. What does it mean?
  • Page 3 line 121- 122: the meaning of the sentence is not clear.
  • Page 3 line 137-138: a figure is needed.
  • The paper examines the addition of stories with the using of devices that increase 2, 3 or 4 times the period of the existing building. This is not clearly written in the text.
  • Page 3-4, lines 144 until 151: some comments are not appropriate as the isolation system is not allocated at the base of the building.
  • Page 5, line 204: story isolation or base isolation. It should be clearly stated.
  • Page 5 line 224: two times of what? Across all the text please use the term story isolation because someone can be confused.
  • The authors should explain the phrase “simulated earthquake” (page 7 line 310).
  • As we can see from the presented modes the building is symmetric. However, the floor plan shown in Figure 1 the building is not symmetric.
  • Page 6, lines 275- 276: how was the scaling made?
  • Table 6: the title must be changed to: ground motions.
  • Figure 7 and Figure 1 correspond to the same building. In Figure 7 the building is symmetric while the building presented in Figure 1 is not symmetric.
  • The authors refer El centro earthquake. However later in table 6 this ground motion is missed.
  • Proposed title: Evaluation ……with story seismic isolation systems
  • English language needs elaboration. The meaning of some sentences is not clear.

Author Response

Thank you for your manuscript review.

I hope that you will consider this paper as suitable publication in your journal.

Reviewer 2 Report

The manuscript presented a story-isolation method to reduce the seismic loads associated with vertical augmentation of existing buildings and to improve seismic performance. The topic is of interest and relevance and fits the journal. However, given the widely known results regarding the story-isolation method, the novelty is limited. The reviewer has the following comments that the authors have to address before the manuscript can be considered for publication.

(1) The current manuscript is more likely constructed in a technical report form because all the analysis results are restricted by the considered design example. Please consider extracting the theoretical model and performing the dimensionless analysis, which can guarantee the wide applicability of the obtained results. 

(2) Considering the well-studied story-isolation technology and widely known effectiveness, the major contribution of this study must be clearly and explicitly explained. The literature review of the story-isolation method should not be ignored. If the only difference between this study and existing ones is the old/existing buildings, unfortunately, the simulation model and method used in this manuscript are the same as the new buildings. 

(3) Apart from the well-known calculation formulas presented in this reporting process, the authors should further explain and illustrate the theoretical contributions of this study in terms of the SPECIAL design methods and performance improvements for existing buildings.

(4) If this is an actual project that has been implemented, please add relevant project information and photos, which will increase the significance of this manuscript.

(5) Figures require much more fidelity, especially Figures 7, 8, and 12. In addition, please expand figure captions so the figures are almost self-explanatory.

Author Response

Thank you for your manuscript review.

I hope that you will consider this paper as suitable publication in your journal.

sincerely.

Reviewer 3 Report

Story-added type apartments have been increasingly introduced, and how to improve the seismic performance of story-added buildings is a vital issue. This paper introduced approaches to reduce the seismic loads and then to improve the seismic performance of story-added buildings. The results show that the application of a seismic isolation system can bring about sufficient reductions in seismic loads. This study falls in the scope of the journal. The manuscript is well written and easy to be understood. In addition, it could provide a reference for the seismic design of similar story-added buildings. Therefore, in my opinion, it could be accepted.

Author Response

Thank you for your manuscript review.

I hope that you will consider this paper as suitable publication in your journal.

Sincerely.

Reviewer 4 Report

In this paper, the seismic performance of the retrofitted storey building with isolation system is studied.There are some tips that need to be modified in the paper. 1. This article is more like a case analysis than a paper. It is only analyzed with finite element software without giving corresponding theoretical research. 2. The abstract is lengthy, unclear and not logical. 3. According to Roehl's articlel, where is the innovation? 4. In the finite element software used in this paper, the hysteretic type should give more specific input parameters used in the model. 5. What about the material properties of the column? The restraint relationship between isolated structure and structural members shall be described. 6. Table II is proposed to be redrawn

Author Response

(The authors gave the same response as above.)

Round 2

Reviewer 1 Report

The authors replied to the reviewers' comments and have done all the modifications/corrections proposed by the reviewers. Hence, I propose the publication of the paper in the present form.

Author Response

Thank you for your review. 

Reviewer 2 Report

1. The current manuscript is more likely constructed in a technical report form because all the analysis results are restricted by the considered design example. Please consider extracting the theoretical model and performing the dimensionless analysis, which can guarantee the wide applicability of the obtained results.

2. Considering the well-studied story-isolation technology and widely known effectiveness, the major contribution of this study must be clearly and explicitly explained. The literature review of the story-isolation method should not be ignored. If the only difference between this study and existing ones is the old/existing buildings, unfortunately, the simulation model and method used in this manuscript are the same as the new buildings.

3. English should be strengthened. Besides, the quality and fidelity of Figures 7, 8, and 12 still need revision.

Author Response

Thank you for your review. 

Reviewer 4 Report

English needs to be strengthened

Author Response

Thank you for your review.

Also, the manuscript was re-edited.